# Transient SARS-CoV-2 RNA-Dependent RNA Polymerase Mutations after Remdesivir Treatment for Chronic COVID-19 in Two Transplant Recipients: Case Report and Intra-Host Viral Genomic Investigation

**DOI:** 10.3390/microorganisms11082096

**Published:** 2023-08-16

**Authors:** Shangxin Yang, Ashrit Multani, Jacob M. Garrigues, Michael S. Oh, Peera Hemarajata, Taylor Burleson, Nicole M. Green, Caspian Oliai, Pryce T. Gaynor, Omer E. Beaird, Drew J. Winston, Christopher S. Seet, Joanna M. Schaenman

**Affiliations:** 1Department of Pathology and Laboratory Medicine, David Geffen School of Medicine at UCLA, Los Angeles, CA 90095, USA; 2Division of Infectious Diseases, Department of Medicine, David Geffen School of Medicine at UCLA, Los Angeles, CA 90095, USA; amultani@mednet.ucla.edu (A.M.); pgaynor@mednet.ucla.edu (P.T.G.); obeaird@mednet.ucla.edu (O.E.B.); 3Public Health Laboratories, Los Angeles County Department of Public Health, Downey, CA 90242, USApeerah@gmail.com (P.H.); tmundt@ph.lacounty.gov (T.B.); nicgreen@ph.lacounty.gov (N.M.G.); 4Division of Hematology-Oncology, Department of Medicine, David Geffen School of Medicine at UCLA, Los Angeles, CA 90095, USA; michaeloh@mednet.ucla.edu (M.S.O.); coliai@mednet.ucla.edu (C.O.); dwinston@mednet.ucla.edu (D.J.W.)

**Keywords:** SARS-CoV-2, Chronic COVID-19, RdRp, nsp3, nsp5, nsp7, nsp12, remdesivir, drug resistance, transplant, intra-host viral evolution

## Abstract

Remdesivir is the first FDA-approved drug for treating severe SARS-CoV-2 infection and targets RNA-dependent RNA polymerase (RdRp) that is required for viral replication. To monitor for the development of mutations that may result in remdesivir resistance during prolonged treatment, we sequenced SARS-CoV-2 specimens collected at different treatment time points in two transplant patients with severe COVID-19. In the first patient, an allogeneic hematopoietic stem cell transplant recipient, a transient RdRp catalytic subunit mutation (nsp12:A449V) was observed that has not previously been associated with remdesivir resistance. As no in vitro study had been conducted to elucidate the phenotypic effect of nsp12:A449V, its clinical significance is unclear. In the second patient, two other transient RdRp mutations were detected: one in the catalytic subunit (nsp12:V166A) and the other in an accessory subunit important for processivity (nsp7:D67N). This is the first case report for a potential link between the nsp12:V166A mutation and remdesivir resistance in vivo, which had only been previously described by in vitro studies. The nsp7:D67N mutation has not previously been associated with remdesivir resistance, and whether it has a phenotypic effect is unknown. Our study revealed SARS-CoV-2 genetic dynamics during remdesivir treatment in transplant recipients that involved mutations in the RdRp complex (nsp7 and nsp12), which may be the result of selective pressure. These results suggest that close monitoring for potential resistance during the course of remdesivir treatment in highly vulnerable patient populations may be beneficial. Development and utilization of diagnostic RdRp genotyping tests may be a future direction for improving the management of chronic COVID-19.

## 1. Introduction

In the community setting, people with chronic underlying conditions, including obesity, hypertension, diabetes mellitus, cardiovascular diseases, and respiratory disorders, are at a higher risk for developing severe COVID-19 [1]. In immunocompromised populations, SARS-CoV-2 infection is associated with prolonged viral shedding, more severe diseases, and generally worse outcomes [2]. Most notably, hematopoietic stem-cell transplantation (HSCT) recipients tend to develop severe and chronic COVID-19, which often leads to poor survival [3]. In solid organ transplant (SOT) patients, particularly non-kidney SOT recipients, COVID-19 was found to be associated with much higher mortality rate (16.2%), compared with 8.3% in non-SOT patients [4]. Therefore, in HSCT and SOT recipients, treatment for COVID-19 is a critical task but often met with great challenges.

Remdesivir is the first FDA-approved drug for treating severe SARS-CoV-2 infection and is indicated for hospitalized COVID-19 patients requiring supplemental oxygen or at high risk of disease progression [5]. As a nucleoside analog, remdesivir inhibits viral replication by targeting the RNA-dependent RNA polymerase (RdRp) complex of SARS-CoV-2, which consists of the catalytic subunit nsp12 and two stabilizing subunits nsp7 and nsp8 [6]. Remdesivir was initially developed for Ebola virus; however, with its potent inhibitory activities against RdRp in many other RNA viruses it has broad-spectrum anti-viral activities on SARS-CoV, MERS, and SARS-CoV-2 [7]. Remdesivir is a monophosphoramidate prodrug of an adenosine analogue, which requires bioactivation through esterase; this limits its bioavailability, and thus it can only be administered intravenously [7,8].

In vitro studies have shown that resistance to remdesivir may emerge through mutations in nsp12, specifically F476L and V553L in SARS-CoV (corresponding to F480 and V557 in SARS-CoV-2, respectively) [6,9]. Notably, global SARS-CoV-2 genomic surveillance has shown very low genetic variation in the region of the *ORF1ab* gene that encodes nsp12, with the only high-frequency non-synonymous mutation resulting in P323L, which is located outside of the catalytic site or key motifs predicted to interfere with the mechanism of action of remdesivir [6]. Interestingly, a study suggested nsp12:P323L may contribute to the emergence of variants with transmission advantages [10]. In clinical use of remdesivir, resistance due to nsp12 mutations has rarely been reported, mostly involving immunocompromised patients with persistent COVID-19. In one post-rituximab B-cell immunodeficient patient, remdesivir treatment failure was associated with nsp12:D484Y mutation [11], occurring in the same key region of the catalytic domain as F480L mutation observed in animal studies of remdesivir resistance [12]. In another case report, two renal transplant patients independently developed nsp12:V792I mutation-associated remdesivir resistance after prolonged remdesivir treatment [13].

Overall, RdRp mutations generally lead to loss of viral fitness, and, therefore, remdesivir-resistance-associated mutations are found very infrequently [8]. However, in the immunocompromised patient population, especially among HSCT and SOT recipients, COVID-19 often tends to be prolonged, and, therefore, it is necessary to monitor for emerging mechanisms of resistance to remdesivir, which remains one of the few treatment options for severe COVID-19. Here, we studied the intra-host evolution of SARS-CoV-2 in a HSCT recipient and a SOT recipient, with the focus on transient mutations in the remdesivir drug target nsp12. With multiple viral sequencing data points carefully analyzed in the clinical context in these two patients, we aimed to discover mutations potentially associated with remdesivir resistance, and to explore possible genetic markers that may be of potential prognostic utility.

## 2. Methods

### 2.1. SARS-CoV-2 PCR

Nasopharyngeal swabs (NPS) were tested using the SimplexaTM COVID-19 Direct Real-Time RT-PCR assay on the LIASON^®^ MDX instrument (DiaSorin Molecular, Cypress, CA, USA), as previously described [14]. This assay targets the SARS-CoV-2 *S* and *ORF1ab* genes. The plasma was tested using the TaqPath COVID-19, Flu A, Flu B Combo Kit (Thermo Fisher Scientific, Waltham, MA, USA), which targets the SARS-CoV-2 *N* and *S* genes (combined into one fluorescent signal channel).

### 2.2. SARS-CoV-2 Sequencing and Mutation Analysis

NPS and plasma samples were sequenced using amplicon-based protocols using either a MiSeq instrument (Illumina, La Jolla, CA, USA) with ARTIC primers (version 3 for Case 1, version 4.1 for Case 2) or the Clear Dx SARS-CoV-2 WGS v3.0 platform (Clear Labs, San Carlos, CA, USA) for Case 2 only. Detailed protocols and quality control criteria were described previously [15,16]. Illumina sequencing data and Clear Labs consensus genomes were analyzed on Terra.bio using the TheiaCoV_Illumina_PE v2.3.0 and TheiaCoV_FASTA v2.3.0 workflows, respectively, (Theiagen Genomics, Highlands Ranch, CO) [17,18,19,20,21] with Wuhan-Hu-1/2019 (GenBank MN908947) used as the reference genome. Consensus genomes used in this study all had greater than 90% reference genome coverage [22].

### 2.3. Viral Load Monitoring

In Case 1, approximately two weeks after the start of remdesivir treatment, plasma and NPS samples were collected every 2–3 days for SARS-CoV-2 PCR. Cycle threshold (Ct) values of the *S* gene and combined *S* and *N* genes were used as proxies to trend viral load in the upper respiratory tract and plasma, respectively. In Case 2, only the NPS samples, collected on day 5, day 22, and day 27 after the start of remdesivir treatment, were tested via PCR and sequenced.

## 3. Results

### 3.1. Clinical Course and Virologic Dynamics

Case 1. A patient in their 50s with acute myeloid leukemia was admitted to our institution in late 2020, for reduced-intensity conditioning and allogeneic HSCT from a matched related donor. Three days prior to HSCT, they developed fevers and nasal congestion and tested positive for COVID-19 infection during conditioning with busulfan and fludarabine. The initial SARS-CoV-2 PCR in NPS showed a Ct value of 17.1 (remdesivir treatment Day −1; Figure 1). Remdesivir (100 mg intravenously administered daily) was started the following day (Day 0) and two units of COVID-19 convalescent plasma (CP) were administered. The patient received allogeneic stem cells on remdesivir Day 3. Remdesivir was discontinued after a standard five-day course; however, on Day 11 there was an acute worsening of hypoxic respiratory failure necessitating high-flow nasal cannula oxygen and ICU transfer. On Day 13, NPS PCR showed a persistently high viral load (Ct = 15.8). A plasma sample collected on Day 14 was also positive (Ct = 30.7), indicating RNAemia consistent with severe COVID-19 [23,24]. Additional units of CP were given on Day 15, and remdesivir was restarted. Two subsequent plasma samples (Day 19 and 23) showed declining viral loads (Ct 31.3 and 33.3, respectively) and PCR of plasma samples taken between Days 26 to 71 remained negative. The viral load in the upper respiratory tract also declined and fluctuated with Ct values between 24.2 and 34 until Day 68, except for a negative result on Day 35 and a “blip” (Ct 19.5) on Day 50, possibly related to high dose glucocorticoids for possible gastrointestinal GvHD (Figure 1). Remdesivir was discontinued on Day 59 due to clinical improvement after a total of 47 days of discontinuous therapy. On Day 71, the patient tested negative via PCR on both NPS and plasma samples and was considered cleared of the virus and discharged.

On Day 81, the patient was readmitted with recurrent acute hypoxic respiratory failure, but SARS-CoV-2 PCR tests on NPS were repeatedly negative. Despite aggressive treatment, the patient expired on Day 101. On postmortem examination via SARS-CoV-2 PCR, a NPS sample tested weakly positive (Ct 33.8); however, two endobronchial lung biopsies were negative. Lung examination demonstrated severe pulmonary fibrosis involving all lobes of the lungs, consistent with post-COVID-19 pulmonary fibrosis, as well as organizing diffuse alveolar damage with potential etiologies, including viral infection, GvHD, or idiopathic pneumonia syndrome following allogeneic HSCT. During the entire course, no other co-infections were presented.

Case 2. A patient in their 70s which had a history of Eisenmenger complex for which they underwent en bloc bilateral orthotopic lung transplantation and orthotopic heart transplantation more than 30 years ago that was complicated by chronic lung allograft dysfunction-bronchiolitis obliterans syndrome (CLAD-BOS). The patient subsequently underwent a single lung transplantation approximately 10 years later that was also complicated by CLAD and pulmonary hypertension for which she required supplemental oxygen administered via nasal cannula (NC). The patient’s immunosuppression regimen consisted of tacrolimus, sirolimus, and prednisone. The patient had received two mRNA COVID-19 vaccine doses approximately two years prior but had not received the bivalent COVID-19 vaccine.

The patient initially presented themselves to another hospital with acute dyspnea, hypoxemia, non-productive cough, rhinorrhea, chills, and fatigue that started the day prior to admission in early 2023. An at-home COVID-19 antigen test resulted positive. In the Emergency Department, the patient was placed on high-flow nasal cannula (HFNC). Remdesivir (200 mg intravenously loading dose followed by 100 mg administered daily) and dexamethasone (6 mg intravenously daily) were started. On Day 5, the patient was transferred to our hospital for ongoing care. A nasopharyngeal COVID-19 PCR was positive with a Ct value of 23.1. The patient received one unit of COVID-19 convalescent plasma (CP). Remdesivir and dexamethasone were continued with ongoing improvement of respiratory status. On Day 12, the patient was discharged home on NC after receiving remdesivir and dexamethasone for 10 days each.

On Day 21, the patient was readmitted to our hospital with ongoing symptoms of dyspnea, productive cough, rhinorrhea, weakness, and fatigue. The patient was again placed on HFNC. On Day 22, NP COVID-19 PCR was positive with a Ct value of 28.7. Another unit of COVID-19 CP was given. Remdesivir and dexamethasone were restarted. On Day 27, NPS COVID PCR was again positive with a Ct value of 17.7. Due to the lack of clinical improvement in conjunction with significantly decreased Ct values (suggestive of increased viral burden) despite targeted therapies, suspicion of antiviral resistance was heightened, and sequencing was performed. On Day 31, the patient elected to transition to comfort-based care and expired. No other co-infections were identified.

### 3.2. Intra-Host Viral Evolution after Remdesivir Treatment

Case 1. The initial NPS sample collected one day prior to remdesivir treatment (Day −1) was sequenced and the virus typed as Pangolin lineage B.1.400 (Clade 20A). The virus from a subsequent NPS sample (Day 23) was also typed as B.1.400 and genetically identical to the virus from the NPS except for an A449V mutation in nsp12 (Figure 1). Sequencing was attempted on the two following plasma samples with higher Ct values but failed to recover sufficient viral genome for analysis. After another six days, the nsp12:A449V mutation was detected in an NPS sample (Day 29) while on remdesivir therapy, which also showed an additional synonymous mutation at genomic position 943 (G943A). However, the nsp12:A449V mutation was short-lived, as five subsequent NPS samples collected between Days 36 to 68 were sequenced and none possessed the nsp12:A449V mutation. Interestingly, examination of individual sequencing reads determined that samples from NPS collected on Day 17 and plasma on Day 14 contained the nsp12:A449V mutation as minor alleles, suggesting intra-host genetic heterogeneity and emergence initially in the plasma. In the approximately two months on remdesivir treatment, aside from the nsp12:A449V mutation found in two samples collected on Days 23 and 29, analysis of consensus sequences detected only one other non-synonymous mutation (nsp5:L24F) in an NPS sample collected on Day 36 (Figure 1) that also contained a synonymous mutation at genomic position 15471 (A15471T).

Case 2. The sequencing of the first NPS sample (Ct 23.1) collected five days after the start of remdesivir treatment revealed a wild-type strain typed as lineage XBB.1.5 (Clade 22F), when the patient was clinically improving. The viral load showed a significant down trend by the Ct value (28.7) of the second PCR result on the NPS collected 17 days later (Day 22) (Figure 2). Interestingly, in this sample with lower viral load collected when the patient was re-admitted due to worsening symptoms, three non-synonymous mutations were detected: nsp3:T820N, nsp7:D67N, and nsp12:V166A. These mutations did not persist and were not found in a subsequent NPS specimen collected five days later (Day 27), despite having significantly higher viral load (Ct 17.7). The high viral load corresponded to a poor clinical response despite remdesivir treatment.

## 4. Discussion

In this study, we observed the emergence of transient mutations in the RdRp complex (nsp7 and nsp12) following remdesivir treatment in two immunocompromised transplant recipients, as well as two other mutations outside of known remdesivir targets (nsp3 and nsp5). In Case 1 (an HSCT patient), we found that although prolonged remdesivir treatment did not result in persistent escape mutants, nsp12:A449V transiently appeared following a standard five-day remdesivir treatment. Nsp12:A449V first emerged in the plasma and then appeared in the upper respiratory tract a week later, suggesting the potential evolution of the virus under selective pressure. The nsp12:A449V mutation has been observed at low frequency in global isolates [6,25], and protein structural analysis showed it resides in the finger domain of the nsp12 subunit, potentially impacting key residues in the RdRp F-motif involved in RNA-template binding [6]. In our case, however, the nsp12:A449V mutant only persisted for approximately two weeks before spontaneous regression, suggesting reduced viral fitness of the mutant. Interestingly, in another reported case, nsp12:A449V mutation was also found in a hypogammaglobulinemic patient with chronic persistent COVID-19 infection approximately 40 days after 17 days of remdesivir treatment [26]. However, in vitro experiments in that study showed remdesivir susceptibility of the isolated viruses after remdesivir treatment, indicating the mutation most likely did not lead to drug resistance. Another mutation observed in Case 1, nsp5:L24F, has not been associated with remdesivir resistance. Nsp5 has been reported to have roles in suppressing antiviral stress granule formation and innate antiviral immunity [27]; however, this mutation also was transient.

In Case 2 (a heart and lung transplant patient), another RdRp mutation, nsp12:V166A, emerged after only five days of remdesivir treatment. Two additional mutations, nsp3:T820N and nsp7:D67N, were also observed at the same time point. Similar to Case 1, all of these mutations were transient and not found in a subsequent sample collected five days later despite a higher viral load. V166 in nsp12 is adjacent to motif D, which plays an important role in closing of the polymerase active site once a nucleoside-triphosphate (NTP) is positioned for incorporation. A recent in vitro study showed that nsp12:V166A emerged in remdesivir resistant strains after prolonged drug selection, but it co-emerged with other mutations including nsp12:S759A and nsp12:C799F [28]; it is unknown whether nsp12:V166A alone can lead to remdesivir resistance. Nsp7 is an accessory subunit of RdRp that interacts with the thumb domain of nsp12 and is important for processivity [29,30], but no mutations in nsp7 have been associated with remdesivir resistance. Nsp3 is a papain-like protease that is important for cleavage and maturation of viral polyproteins, and though T820 might be important for proper functionality of the protease domain [31], nsp3 has not been associated with remdesivir resistance. These findings show that multiple transient mutations were present in Day 22 specimen, two of which were present in the RdRp complex targeted by remdesivir. However, it is unknown whether any of these mutations had an effect on remdesivir susceptibility or may have contributed to this patient’s demise.

Limitations of this report are that our observations are from only two patients, and that convalescent plasma therapy and donor cell engraftment may have contributed to remdesivir-independent clearance of the mutant, thus confounding interpretation of steady-state viral fitness. Second, except for the nsp12:A449V mutation, we only investigated the major alleles that are present in consensus sequences; minor alleles that correspond to known remdesivir targets may be present but unaccounted for. Third, in vitro phenotypic testing for remdesivir resistance was not performed in this study due to limited resources available for viral culture. Another limitation is that in both cases, the mutations were transient, and did not persist to the end of the treatment, thus confounding the interpretation of their true clinical significance. It is likely that these mutations may only be precursors for true remdesivir resistance or a biological signal predicting a poor prognosis. Further studies are needed to expand our understanding of virologic dynamics under prolonged remdesivir treatment, the role of prolonged remdesivir therapy in immunocompromised patients, and of the broader incidence and clinical relevance of the mutations observed in these cases.

In conclusion, our study revealed SARS-CoV-2 genetic dynamics during remdesivir treatment in transplant recipients and highlighted the possibilities of mutations in remdesivir drug target RdRp, which may require close monitoring during the course of treatment in these highly vulnerable patient populations. Development and utilization of diagnostic RdRp genotyping tests, similar to other anti-viral resistance tests, may be a future direction for improving the management of chronic COVID-19.

## Figures and Tables

**Figure 1 microorganisms-11-02096-f001:**
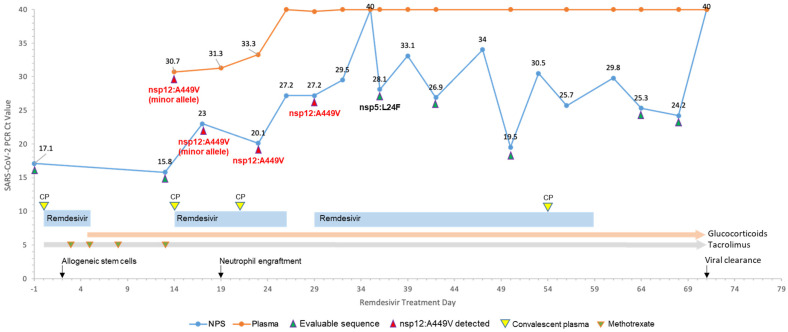
Timeline for treatment and SARS-CoV-2 virological dynamics in Case 1. Timeline for Case 1 (allogeneic HSCT), COVID-19 treatment and SARS-CoV-2 PCR Ct trends. Arrows indicate samples that were sequenced with sufficient genome coverage; red arrows indicate sequences with the nsp12:A449V mutation. Yellow arrows indicate convalescent plasma (CP) treatment. Tacrolimus, methotrexate, and glucocorticoids were given per institutional standards for graft-versus-host disease (GvHD) prophylaxis. Glucocorticoid doses were adjusted throughout treatment for GvHD prophylaxis, empirical treatment of GvHD, and/or treatment of severe COVID-19.

**Figure 2 microorganisms-11-02096-f002:**
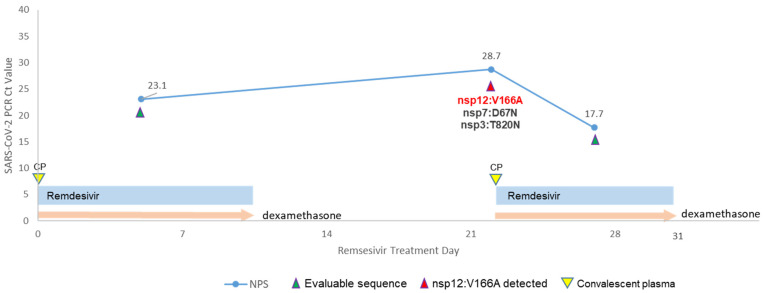
Timeline for treatment and SARS-CoV-2 virological dynamics in Case 2. Timeline for Case 2 (heart and lung transplant), COVID-19 treatment and SARS-CoV-2 PCR Ct trends. Arrows indicate samples that were sequenced with sufficient genome coverage; red arrows indicate specimens containing nsp3:T820N, nsp7:D67N, and nsp12:V166A mutations. Yellow arrows indicate convalescent plasma (CP) treatment.

## Data Availability

Consensus genome sequences were deposited to GISAID and GenBank; raw sequencing data were submitted to SRA. Accession numbers for each sample are listed in Table A1.

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
