# Peer review of "Transient SARS-CoV-2 RNA-Dependent RNA Polymerase Mutations after Remdesivir Treatment for Chronic COVID-19 in Two Transplant Recipients: Case Report and Intra-Host Viral Genomic Investigation"

_microorganisms, 2023, doi:10.3390/microorganisms11082096_

Round 1

Reviewer 1 Report

The present case report evaluates the transient SARS-CoV-2 RNA-dependent RNA polymerase mutations after remdesivir treatment for Chronic COVID-19 in two transplant recipients. The topic is relevant and up-to-date, but certain deficiencies identified in both content and form need to be addressed based on the specific recommendations below:

The information section of the introduction is organised as an overly long paragraph, which reduces readability and comprehension. Please reorganise it into shorter paragraphs, which will be more logical and easier to understand.

The introduction starts too abruptly/targeted, omitting the general date presentation part. It is necessary to briefly present data on COVID-19, risk assessment, associated diseases with implications at molecular level and in the progression of infection, role of CD147-spike protein. I suggest you check and consult: PMID: 36406478 and PMID: 34863742. 

The aim of the paper should be presented separately in the last paragraph of the introduction and should be approached from the perspective of describing the contribution to the field under review and the elements of scientific novelty presented. 

It is necessary to detail the pharmacological properties of RDV in order to better understand the mechanisms of interaction between virus and drug and to what extent these may lead to the emergence of resistance. I suggest you check and consult: PMID: 35131656

Author Response

Reviewer-1

The present case report evaluates the transient SARS-CoV-2 RNA-dependent RNA polymerase mutations after remdesivir treatment for Chronic COVID-19 in two transplant recipients. The topic is relevant and up-to-date, but certain deficiencies identified in both content and form need to be addressed based on the specific recommendations below:

  1. The information section of the introduction is organized as an overly long paragraph, which reduces readability and comprehension. Please reorganize it into shorter paragraphs, which will be more logical and easier to understand.

Response: thank you for your suggestion. We have revised the Introduction section to make it more logical and organized (Lines 45-91).

  1. The introduction starts too abruptly/targeted, omitting the general date presentation part. It is necessary to briefly present data on COVID-19, risk assessment, associated diseases with implications at molecular level and in the progression of infection, role of CD147-spike protein. I suggest you check and consult: PMID: 36406478 and PMID: 34863742. 

Response: thank you for your suggestion. We have revised the Introduction section and added the 1st review paper (PMID: 36406478) (Lines 45-47). We feel this study is not related to CD147-spike protein and therefore opted out citing the 2nd paper.

  1. The aim of the paper should be presented separately in the last paragraph of the introduction and should be approached from the perspective of describing the contribution to the field under review and the elements of scientific novelty presented. 

Response: thank you for your suggestion. We have revised the Introduction section and added the aims in the last paragraph of the Introduction section (Lines 88-91).

  1. It is necessary to detail the pharmacological properties of RDV in order to better understand the mechanisms of interaction between virus and drug and to what extent these may lead to the emergence of resistance. I suggest you check and consult: PMID: 35131656

Response: thank you for your suggestion. We have revised the Introduction section and added this wonderful review article to further describe the pharmacological properties of RDV (Lines 60-64).

Reviewer 2 Report

In this study, the author described two COVID-19 cases who developed mutations during remdesivir therapy. The first case developed a transient RdRp catalytic subunit mutation (nsp12:A449V), and the second case two other transient RdRp mutations were detected: one within the catalytic subunit  (nsp12:V166A) and the other in an accessory subunit important for processivity (nsp7:D67N). 

Major points

1) IRB approval should be taken , even if the patients' consent is not required.

2) The author should submit the Seq data to Genbank and provide the NCBI ID in the manuscript

3) The lab and clinical data are missing in those two cases, and should be monitored overtime with viral load to see the effect of mutation emergences on these parameters

4) Did the authors have any microbiological data on these two cases? I mean did mutations affect the growth of secondary bacteria infection?

Minor language editing

Author Response

Reviewer 2

In this study, the author described two COVID-19 cases who developed mutations during remdesivir therapy. The first case developed a transient RdRp catalytic subunit mutation (nsp12:A449V), and the second case two other transient RdRp mutations were detected: one within the catalytic subunit  (nsp12:V166A) and the other in an accessory subunit important for processivity (nsp7:D67N). 

Major points

  1. IRB approval should be taken, even if the patients' consent is not required.

Response: thank you for your suggestion. Since this study is within the scope of public health surveillance for tracking SARS-CoV-2 variant conducted by the Los Angeles County Department of Public Health (LACDPH), and IRB approval is not required. We have added this in the ethical statement (Line 303).

  1. The author should submit the Seq data to Genbank and provide the NCBI ID in the manuscript

Response: thank you for your suggestion. We have uploaded all the sequencing data to Genbank and provided Table 1 summarizing all the accession numbers (Line 413).

  1. The lab and clinical data are missing in those two cases, and should be monitored overtime with viral load to see the effect of mutation emergences on these parameters

Response: thank you for your suggestion. However, we believe we have provided full description of the 2 cases and didn’t leave any important clinical or lab information. In addition, we also use the PCR Ct value as a surrogate marker for viral load trending, which is explained in the Method section. Please specify a particular lab result or clinical data that should be included if warranted.

  1. Did the authors have any microbiological data on these two cases? I mean did mutations affect the growth of secondary bacterial infection?

Response: thank you for your suggestion. In both cases, no other significant microbiological findings were present. These were pure COVID-19 infections. We added this information in Line 164 and Line 195.

Round 2

Reviewer 1 Report

The authors have significantly improved the manuscript based on the suggestions received.

Reviewer 2 Report

No further comments

Language is fine